# TreeSNNs: Temporal Resolution Ensembled SNNs for Neuromorphic Action Recognition

## Abstract

Spiking Neural Networks (SNNs) are energy-efficient due to sparse and asynchronous event processing, but their accuracy often lags behind that of conventional ANN deep learning models. Prior research has largely focused on novel SNN architectures and training methods, with continuous event streams typically binned into frames using either fixed event counts or fixed time intervals. In this paper, we observe that different motions exhibit distinct temporal dynamics and may be best captured at different temporal/event resolutions. Building on this insight, we propose TreeSNNs, an ensemble framework where model diversity is expressed via multiple event temporal resolutions. We utilize the Fano factor as a metric to quantify temporal dynamics and guide the selection of such a diverse set of temporal resolutions tailored to a given dataset. Individual SNNs trained at these resolutions are then aggregated through ensembling to improve recognition accuracy. Experiments on three neuromorphic action datasets—DVS Gesture, SL-Animals DVS, and the challenging THU$^{E\text{-}ACT}$-50 CHL—show that TreeSNNs consistently outperform baselines, improving accuracy by $1.05\%$–$6.8\%$.

## 1 Introduction

Spiking Neural Networks (SNNs) have emerged as a promising technology owing to their energy efficiency, which is achieved through the processing of discrete spikes. Moreover, SNNs are inherently sequential, allowing them to capture information across multiple timesteps and making them well-suited for modeling temporally evolving data. However, despite these advantages, a substantial performance gap persists compared to conventional deep learning models (ANNs), on fundamental tasks such as object classification, tracking, and action recognition (Nunes et al., 2022; Deng et al., 2020; Hao et al., 2023). This gap is largely attributed to the the non-differentiability of spikes, the absence of effective training strategies, and the limited availability of mature libraries and development tools for SNNs.

To address these challenges, numerous studies have focused on improving training mechanisms. For instance, Shrestha & Orchard (2018); Rathi & Roy (2021); Liu et al. (2025) proposed training strategies that enhance gradient flow and mitigate spike non-differentiability, while Li et al. (2023) introduces techniques such as dynamic confidence for early decision-making, eliminating redundant timesteps and achieves low-latency processing. More recently, **?** explored ensemble learning, where ensembles of models across timesteps, guided by subnetworks, further improved performance. However, this ensemble approach does not exploit diversity in how events are represented and presented to SNNs, limiting its ability to capture event-stream dynamics at differentiated temporal resolutions.

Traditionally, event frames for SNNs are generated by temporally binning the event stream, either uniformly by time (Fischer & Milford, 2022) or by *events* (Maqueda et al., 2018)—this latter event count-based approach is better suited to capture the time-varying motion dynamics of underlying physical events. However, even *uniform* event binning has a limitation: a single binning resolution (denoted by $N_f$, the total number of bins) can translate to widely varying time resolution across classes that have very different motion dynamics (e.g., hand waving vs. rapid air-drumming), making it difficult to incorporate *micro-motion artifacts* in the neural learning process. Figure 1 illustrates two actions from two gesture datasets. With timesteps $T = 5$ and $N_f = 5$, both actions display similar motion patterns across timesteps, an artifact of uniform event binning. When $N_f = 20$ or $N_f = 40$, the event stream is divided into 20 or 40 bins, respectively. To align with the desired

processing length ($T = 5$ in this case), we uniformly sample five frames from these finer partitions, which more clearly capture the temporal evolution of actions compared to $N_f = 5$. This observation highlights that different actions require different event bin resolutions, rather than a single $N_f$ value, to be effectively represented and processed by SNNs.

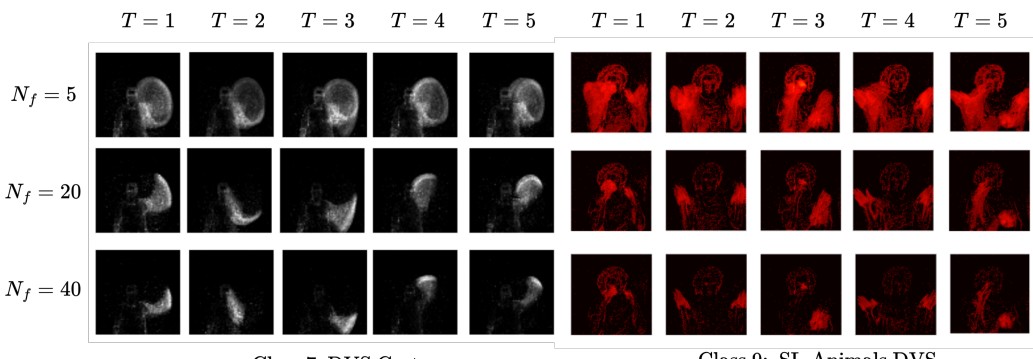

Figure 1: of motions across timesteps for different $N_f$.

Our key contribution in this paper is to extend the principle of *ensemble learning* to SNN architectures via the concept of **temporal ensembling**–i.e., characterizing model diversity in terms of varying temporal/event resolutions. Our proposed framework, called *TreeSNNs* (**T**emporal **Re**solution **E**nsembled SNNs) goes beyond extant ANN-based notions of model diversity, generated either through randomized seeds Zhi et al. (2023) or spatial resolution Jiang et al. (2019), to develop the SNN-specific concept of *temporal model diversity*. To systematically develop the *TreeSNNs* framework, we first introduce the Fano factor, a statistical measure commonly used in neuroscience to model neural spiking dynamics (Rajdl et al., 2020), as an effective metric for quantifying the temporal dynamics of an event stream. Based on the Fano factor, we then design an algorithm to select a set of $N_f$ values that can maximize the distinction across different action classes, allowing the composite SNN model to capture motion segments at diverse temporal resolutions. Our key contributions are as follows:

- Based on the analysis of event binning schemes, we proposed to enhance SNN-based ensemble learning to exploit the temporal diversity of event streams by having different models operate at different *temporal event resolution*.

- We introduced the Fano factor to quantify temporal dynamics in event streams and designed an algorithm to select a set of event bin resolutions that maximize inter-class temporal differences. The selected $N_f$ values are then used to generate frame and train individual SNNs, whose outputs are ensembled for final classification.

- We conducted extensive experiments on three neuromorphic action datasets. Our approach improves accuracy over SOTA models by *1.05% (T=5)* on DVS-Gesture and *2.05% (T=10)* SL-Animals DVS, and achieves a substantial *6.8% (T=10)* improvement over the baseline on the challenging THU[E-ACT]-50 CHL dataset. We also analyzed the relationship between ensemble size and accuracy, and visually revealed how temporal ensembling contributes to performance gains.

## 2 RELATED WORKS

We first review recent advances in improving SNN performance, both through general architectural design and training mechanisms, and via novel representations and processing techniques that explicitly consider the temporal dynamics of event streams.

### 2.1 IMPROVING SNNS VIA ARCHITECTURAL DESIGN AND TRAINING MECHANISMS

To improve the performance of SNNs, recent research has focused on both architectural innovations and training mechanisms. On the architectural side, transformer-inspired designs have gained attention, such as Spikeformer (Zhou et al.), which introduces spiking self-attention modules, and

SpikingResformer (Shi et al., 2024), which integrates dual spiking self-attention with a ResNet backbone to boost performance. Training approaches can be broadly divided into direct backpropagation-based methods and ANN-to-SNN conversion. Direct training strategies, such as Diet-SNN (Rathi & Roy, 2021), have shown significant progress, yet conventional backpropagation remains challenged by the temporal dynamics of spikes and their non-differentiability. To address this, Shrestha & Orchard (2018) proposed a spike-timing-aware backpropagation algorithm, while Liu et al. (2025) propose to dynamically adapt surrogate functions based on membrane potential distributions. By contrast, ANN-to-SNN conversion bypasses the differentiability problem but often incurs quantization-induced accuracy loss and requires longer timesteps. Recent works (e.g., Bu et al.; Wang et al. (2023)) alleviate these issues by optimizing activation functions, thereby reducing conversion error, lowering timestep requirements, and improving robustness.

### 2.2 Improving SNN via Temporal Dynamics of Event Streams

SNNs are inherently adept at modeling temporal dynamics, while neuromorphic event streams naturally encode rich temporal information. To fully exploit this synergy, recent research has increasingly focused on optimizing the temporal processing of SNNs to achieve higher accuracy with lower latency. For example, Li et al. (2023) introduced a dynamic confidence strategy that skips redundant inference timesteps to reduce latency. Ding et al. (2024) investigated heterogeneous temporal scales within SNN layers, enabling different stages of a single SNN model to operate at different timesteps. To improve generalization across temporal resolutions, Du et al. proposed Mixed-Timestep Training. Additiionally, Ding et al. (2025) re-framed SNN inference as an ensemble of subnetworks across timesteps, stabilizing predictions through membrane potential smoothing. However, the idea of ensembling individual SNNs that specialize in distinct temporal dynamics remains underexplored.

### 2.3 Enhancing Temporal Representation with Event Binning

Recent works have explored novel *adaptive* event representations that move beyond the conventional approaches of uniform temporal binning (Fischer & Milford, 2022; Rebecq et al., 2016) (fixed time intervals) or uniform event count binning (Maqueda et al., 2018; Zhu et al., 2022) (fixed number of events). For instance, Sen et al. (2024) proposed an adaptive slicing strategy that triggers a new slice when the standard deviation of event polarities, computed via a moving average, exceeds a threshold. Similarly, Cao et al. (2024) introduced SpikeSlicer, which leverages output spike timings of an SNN to guide the slicing process. Although adaptive binning yields more temporally relevant motion representations, existing methods typically rely on a single SNN to process all bins, across multiple temporal resolutions. It seems natural and intuitive to instead explore an ensemble approach, where each individual model can *specialize* in distinct action patterns, thereby enabling the ensemble to achieve superior performance by stronger generalization across a range of timescales.

## 3 Methodology

We now introduce *TreeSNNs*, an ensemble approach for neuromorphic action recognition that leverages multi-resolution event binning. We first show how multi-resolution binning captures diverse temporal dynamics, then present an algorithm for selecting a set of resolutions that maximize temporal discriminability, and finally ensemble the corresponding models for improved classification.

### 3.1 Multi-resolution for Effective Temporal Dynamic Capture

**Event Stream Binning and Framing.** Asynchronous event streams captured by DVS cameras must be pre-processed to expose their temporal dynamics before being fed into SNNs. This typically involves binning and framing. In binning, the stream is divided along the time axis into 3D event bins. These bins are then aggregated into 2D event frames through framing, which are subsequently processed by SNNs across timesteps $T$ (where $T$ usually equals the number of frames).

There are two primary binning methods: uniform time-binning and uniform event-binning. In uniform time-binning, the event stream is divided into bins of equal duration. Figure 2 (red bins) shows an example where a sample is split into $T = 3$ equal time bins. Since the division depends only on time, bins may contain vastly different numbers of events. Fast motion can overload bins with dense

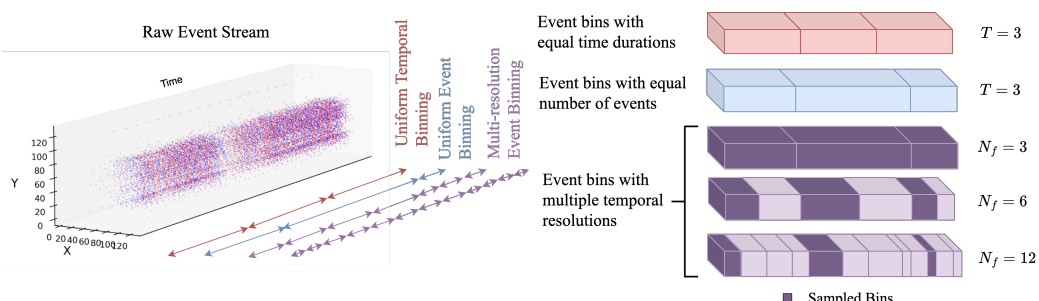

Figure 2: Uniform time, uniform event and multi-temporal resolution binning.

events, while slow movements may produce sparse bins. This imbalance can result in information loss or noise, reducing the effectiveness of SNN processing. In uniform event-binning, the stream is divided such that each bin contains the same number of events (e.g., $T = 3$ bins with equal event counts, shown in blue in Figure 2). Unlike time-binning, the temporal span of each bin is variable: low activity leads to long time windows that blur fine dynamics, while high activity produces very short bins that overemphasize (generate larger number of bins of) segments of fast motion. Because this differentiated sampling is *independent of the underlying saliency of that motion segment* (e.g., a fast vertical motion of a user's hand may be overrepresented, compared to the slow but more discriminative rotation gesture segment), it can lead to a loss in the fidelity of capturing key motion artifacts and reduced classification accuracy.

To address these limitations, we propose leveraging *multiple temporal resolutions* for binning and framing. Specifically, we first apply uniform event-binning to split the stream, but instead of a single resolution, we divide each sample into multiple resolutions with different numbers of bins ($N_f$). These bins are then aggregated into frames. Since different $N_f$ values produce varying numbers of frames, we uniformly sample across time to obtain a fixed number of frames $T$ for each resolution. This ensures consistency across resolutions while still capturing the full motion trajectory. Figure 2 (violet bins) shows an example with $N_f = 3, 6, 12$ and $T = 3$.

**Temporal Dynamics: Dynamic vs. Static Dataset.** Compared to uniform time-binning, multiple temporal resolutions provide balanced representation of different motion segments while still capturing such motion across diverse temporal scales. Unlike uniform event-binning, where a fixed bin size risks overemphasizing periods of fast motion (or underemphasizing periods of slow motion), multiple $N_f$ introduces temporal diversity by preserving both fine- and coarse-grained dynamics, making it especially effective for dynamic, motion-based datasets.

To validate this, we trained three SNN classifiers (using VGG9 architecture) on the DVS-Gesture dataset (a dynamic neuromorphic gesture dataset, details can be found in Section 4.1) with $N_f = 5, 20, 40$, respectively. Figure 3 (upper row) illustrates the confusion matrix for each $N_f$. We can observe that while the three models achieve relatively comparable overall accuracy, they specialize in different classes and thus complement each other. For example, Class 4 reaches 92% accuracy with $N_f = 40$, while the coarser temporal resolution provided by $N_f = \{5, 20\}$ results in significantly higher classification error. More concretely, when $N_f = 5$, Class 4 (clockwise rotation) is frequently confused with Class 5 (counter-clockwise rotation), due to insufficient temporal resolution. Conversely, the frames with $N_f = 40$ better capture the finer-grained motion dynamics, promoting higher accuracy. Clearly, different action classes benefit from different temporal resolutions.

To further validate that the benefits of multi-resolution binning arise from the underlying non-uniform temporal dynamics, we repeated the experiment on the CIFAR-10-DVS dataset. Unlike the DVS-Gesture dataset, CIFAR-10-DVS is generated by applying slight perturbations to static CIFAR-10 images and recording them with an event camera, and therefore lacks intrinsic variability in the motion dynamics. As shown in Figure 3 (lower row), classifiers trained with different $N_f$ values learn largely similar representations: all models perform worst on Class 3 and best on Classes 1 and 9. There is almost no instance where one model complements another, since the dataset does not encode meaningful temporal evolution.

## 3.2 FANO FACTOR BASED TEMPORAL RESOLUTION SELECTION

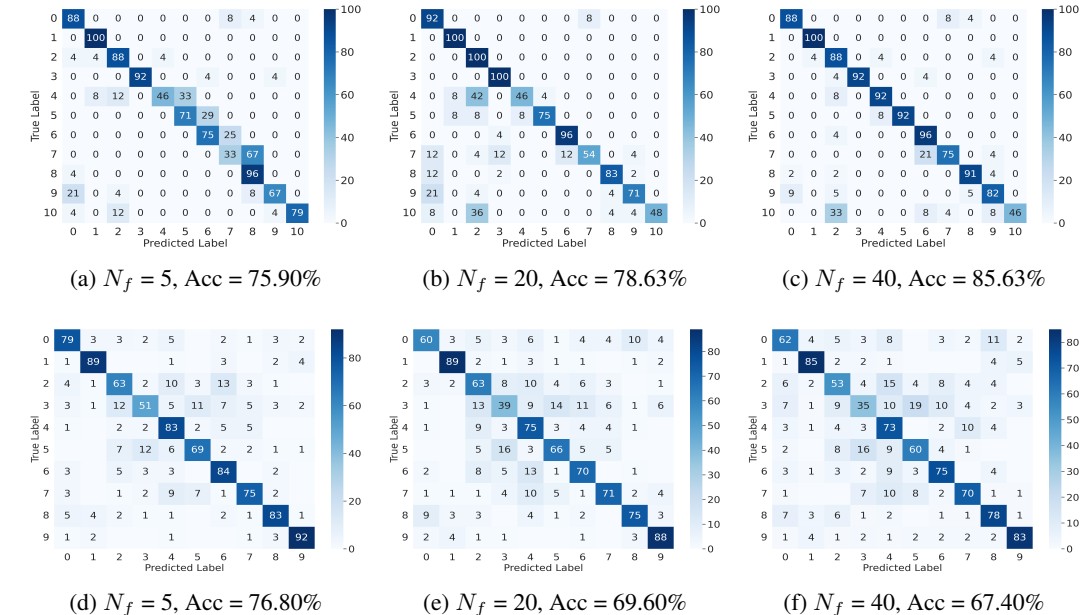

(a) $N_f = 5$, Acc = 75.90%  (b) $N_f = 20$, Acc = 78.63%  (c) $N_f = 40$, Acc = 85.63%

(d) $N_f = 5$, Acc = 76.80%  (e) $N_f = 20$, Acc = 69.60%  (f) $N_f = 40$, Acc = 67.40%

Figure 3: Comparison of confusion matrix for DVS-Gesture (upper row) and CIFAR-10-DVS (lower row) across different $N_f$ values.

To leverage the temporal dynamics captured by our multi-temporal resolution binning, we develop an algorithm to select a set of $N_f$ values that provide optimal diversity for ensemble SNN learning. As illustrated in Figure 4, the algorithm design process is guided by three objectives. First, we quantify the temporal dynamics of event frames using the Fano-factor, which measures the variability of event count at different temporal resolutions. Second, we compute the inter-class distances for each $N_f$ to represent how well an individual resolution choice can differentiate between different action classes. Finally, we select the subset of $N_f$ values that maximizes both intra-and-inter class diversity across resolutions. We now detail the approach for each of these objectives.

*Objective 1: Quantifying temporal dynamics.* To identify the optimal set of $N_f$ values, we first need to quantify temporal dynamics. For the aggregated event data within a 2D frame, temporal dynamics can be reflected by the variability of event counts across pixels. Drawing inspiration from neuroscience, where neural spiking variability is commonly measured using the Fano factor (Rajdl et al., 2020), we adopt it as our metric for quantifying the temporal dynamics. Specifically, for each event frame, we compute the Fano factor as the ratio of variance to mean of its event counts. To do this, each event frame is flattened into a one-dimensional vector $E_{\text{flattened}}$, and the Fano factor is calculated as:

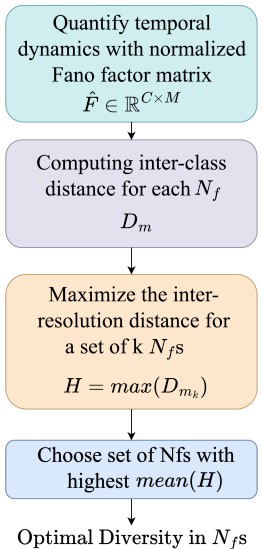

Figure 4: Flow of optimial diversity for $N_f$ selection.

$$\text{Fano} = \frac{\text{Var}[E_{flattened}]}{\text{Mean}[E_{flattened}]}, \quad E_{flattened} \in \mathbb{R}^{1 \times HW} \tag{1}$$

where $H$ and $W$ denote the spatial dimensions of the frame. For a given $N_f$, we compute these Fano factors for every class to represent the class-specific motion dynamics. This is done by taking the mean of the Fano factors across all samples and frames within each class. As a result, each $N_f$

produces a $C \times 1$ vector, where $C$ is the number of classes. For a list of $M$ values of $N_f$, we obtain the Fano factor matrix:

$$F \in \mathbb{R}^{C \times M}, \quad F[c, m] = \text{Fano factor of class } c \text{ at the } m^{th} \text{ temporal resolution of } N_f \quad (2)$$

This matrix compactly encodes the variability of motion dynamics across classes and temporal resolutions, serving as the foundation for selecting a diverse set of $N_f$ values.

*Objective 2: Computing inter-class distance for each $N_f$.* The Fano factor values computed above quantify the temporal dynamics of each class. To improve classification accuracy, it is important to consider the inter-class differences in these dynamics. Accordingly, we compute the inter-class distance of Fano factors for each $N_f$. The distance between class pairs reflects how distinct the temporal dynamics are at a given resolution, indicating the discriminative power of that $N_f$. First, we normalize the Fano factor matrix to $\hat{F}[j, m]$ to ensure comparability across different $N_f$. We then compute the inter-class distance matrix $D_m$, which captures the distinctions between classes:

$$D_m[i, j] = \left| \hat{F}[i, m] - \hat{F}[j, m] \right|, \quad (3)$$

where $\hat{F}[i, m]$ denotes the Fano factor of class $i$ at the $m^{th}$ temporal resolution.

*Objective 3: Maximizing inter-resolution distance.* After computing the inter-class distances for each temporal resolution, the next step is to select a set of $N_f$ values for ensembling. If the target ensemble size is $N_{\text{models}}$, we choose the same number of $N_f$ values, forming a candidate set. From the total pool of $M$ possible $N_f$ values, we exhaustively evaluate all $\binom{M}{N_{\text{models}}}$ combinations of $N_f$. For each candidate set, we apply element-wise maximization across the corresponding inter-class distance matrices $D_m$, producing a matrix $H$, as follows:

$$H[i, j] = \max_{k \in m_1, \ldots, m_{N_{\text{models}}}} D_{m_k}[i, j]. \quad (4)$$

The maximization operation captures the complementary temporal dynamics offered by different $N_f$ values within the candidate set. We then compute the mean of all entries in $H$ to obtain a score for the set. This procedure is repeated for all candidate sets, and the one with the highest score is regarded as providing the most complementary class separations and is therefore selected as the final set of $N_f$ values for ensembling.

$$\text{score} = \frac{1}{C^2} \sum_{i=1}^{C} \sum_{j=1}^{C} H[i, j], \quad (5)$$

### 3.3 Temporal Ensemble on SNNs

Using the selected bin resolutions, we train separate SNN models, each corresponding to a specific $N_f$. This allows each model to specialize in capturing event dynamics at its own temporal resolution. When $N_f > T$, the number of bins exceeds the model's timesteps, so we uniformly sample $T$ bins from the $N_f$ sequence to cover the entire motion trajectory.

$$s_i = \left\lfloor \frac{i \cdot N_f}{T} \right\rfloor, \quad i = 0, 1, \ldots, T - 1$$

where $N_f$ = total number of bins in a sample, $T$ = number of timesteps required by the SNN, $s_i$ = index of the selected bin.

The sampled bins for each $N_f$ are then used to train an individual SNN. At inference, we ensemble all models for aggregated predictions. Instead of uniform averaging, we apply a weighted combination, where each model's weight is defined as $W_n = 1 - (N_{\text{miss}}/N_{\text{samples}})$, with $N_{\text{miss}}$ denoting its misclassified samples (during training) and $N_{samples}$ denoting the total number of samples. For each class $c$, we compute the weighted score $S_c$ by aggregating model outputs with their respective weights, and the final prediction is obtained as $\arg\max_c S_c$.

# 4 EXPERIMENTS

## 4.1 DATASETS AND IMPLEMENTATIONS

We experimented on three neuromorphic action datasets.

**DVS-Gesture (Amir et al., 2017).** This is a standard neuromorphic vision dataset recorded with a DVS128 event camera at 128×128 resolution. It contains 1,464 samples of 11 hand and arm gestures performed by 29 subjects under three illumination conditions, typically split into 1,176 training and 288 testing samples. Following prior work Ding et al. (2025), we train this dataset on two SNN architectures: SpikingResformer-Ti and Spiking-VGG9, with T=5 timesteps.

**SL-Animals DVS (Vasudevan et al., 2022).** It is an event-based sign language dataset ($\approx$1,100 samples spanning 19 isolated animal-related signs) performed by 58 subjects, collected using the DVS128 event camera. We adopt the SpikingResformer-Ti architecture and train it with T=10 timesteps. The baseline methods use uniform event binning.

**THU$^{\text{E-ACT}}$-50 CHL (Gao et al., 2023).** It is a large-scale and challenging neuromorphic action recognition dataset, which comprises 50 action categories captured using a DAVIS346 event camera at a resolution of 346×260 under varied conditions, making it one of the most comprehensive resources for evaluating neuromorphic models. While no prior work has applied SNNs to this dataset, Wang et al. (2024a) reported 49.5% accuracy using the Event-based Spatio-Temporal Transformer (ESTF), an ANN-based architecture. For our SNN baseline, we implemented SpikingResformer-Ti with T=10 timesteps and uniform event binning, achieving an accuracy of 67.28%.

We trained all Spiking-VGG9 and SpikingResformer-Ti models using an NVIDIA V100 GPU with 16 GB memory. The inputs from each dataset were normalized and resized to a spatial resolution of $64 \times 64$ with three channels. Training was performed with a batch size of 8 using the AdamW optimizer with a weight decay of 0.05 and an initial learning rate of $1 \times 10^{-4}$, scheduled using a cosine learning-rate decay. The models were initialized with a manual seed (12450) and trained for 300 epochs.

## 4.2 BASELINES

We compare TreeSNNs with state-of-the-art SNN models, including SNN-EL (Ding et al., 2025) based on VGG9, SDT (Yao et al., 2023) based on Spiking Transformer, and SNN-EL (Ding et al., 2025) based on SpikingResformer-Ti on DVS-Gesture dataset; EventRPG (Sun et al., 2024) based on ResNet-18, and SpikeHAR++ (Lin et al., 2024) based on Spiking Transformer on SL Animal DVS dataset; ESTF (Wang et al., 2024a), TSM (Wang et al., 2024a) for THU$^{\text{E-ACT}}$-50 CHL dataset; as well as SpikingResformer (Shi et al., 2024) based on SpikingResformer-Ti for all datasets.

## 4.3 MAIN RESULTS

Table 1 presents the overall results. On the DVS-Gesture dataset, SNN-EL (Ding et al., 2025) achieves the current state-of-the-art with 94.44% accuracy using SpikingResformer-Ti at $T = 5$ timesteps. Under the same setting, our proposed TreeSNNs framework attains 95.49%, setting a new benchmark. For the VGG9 architecture, TreeSNNs also surpasses the existing best result of 93.23%. Similarly, on the SL-Animals DVS dataset, TreeSNNs achieves the highest accuracy, exceeding prior works by 2%. This improvement demonstrates the effectiveness of TreeSNNs.

To date, no SNN-based classification results have been reported on THU$^{\text{E-ACT}}$-50 CHL. ANN-based methods such as ESTF (Wang et al., 2024b;a) and TSM (Lin et al., 2019) achieved 49.5% and 49.07% accuracy, respectively. Using the SpikingResformer-Ti architecture, we achieve 67.28% accuracy with a single SNN model. With an ensemble of $N_f = 10, 15, 30, 40, 100$, TreeSNNs achieved a significant 6.8% higher accuracy (74.08%), surpassing all individual models and prior ANN-based methods.

## 4.4 ABLATION STUDY

We conducted ablation studies using SpikingResformer-Ti on three datasets to evaluate the effectiveness of our $N_f$ selection strategy, by comparing it against randomized selection. For a given

Table 1: Comparison with state-of-the-art methods on neuromorphic action datasets. (*: implemented by the authors with the available GitHub code.)

| Dataset | Method | Architecture | T | Accuracy (%) |
|---|---|---|---|---|
| DVS-Gesture | SNN-EL (Ding et al., 2025) | VGG9 | 4 | 93.23 |
| | SDT (Yao et al., 2023) | Spiking Transformer-2-256 | 5 | 92.24 |
| | SNN-EL (Ding et al., 2025) | SpikingResformer-Ti | 5 | 94.44 |
| | SpikingResformer (Shi et al., 2024) | SpikingResformer-Ti | 5 | 89.24* |
| | **TreeSNNs (Ours)** | VGG9 | 5 | **93.75** |
| | | SpikingResformer-Ti | 5 | **95.49** |
| SL-Animals DVS | EventRPG (Sun et al., 2024) | SEW ResNet-18 | 16 | 91.59 |
| | SpikeHAR++ (Lin et al., 2024) | Spiking Transformer-6-512 | 10 | 91.93 |
| | SpikingResformer (Shi et al., 2024) | SpikingResformer-Ti | 10 | 91.35* |
| | **TreeSNNs (Ours)** | SpikingResformer-Ti | 10 | **93.98** |
| THU$^{\text{E-ACT}}$-50 CHL | ESTF (Wang et al., 2024a) | SF + TF + FusionFormer | – | 49.50 |
| | TSM (Wang et al., 2024a) | TSM ResNet-50 | – | 49.07 |
| | SpikingResformer (Shi et al., 2024) | SpikingResformer-Ti | 10 | 67.28* |
| | **TreeSNNs (Ours)** | SpikingResformer-Ti | 10 | **74.08** |

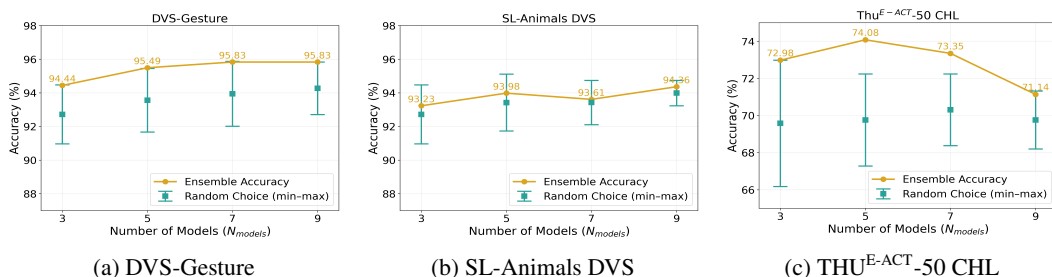

(a) DVS-Gesture  (b) SL-Animals DVS  (c) THU$^{\text{E-ACT}}$-50 CHL

Figure 5: Ablation study: comparison between random choice and fano-factor based selection.

target ensemble size, $N_{\text{models}}$, we randomly sampled $N_{\text{models}}$ $N_f$ values and repeated the process 10 times to capture performance variability.

Results are summarized in Figure 5. On DVS Gesture, TreeSNNs performed on par with the best random selections across different $N_{\text{models}}$ (Figure 5a). On THU$^{\text{E-ACT}}$-50 CHL, our method consistently outperformed random selection, especially for $N_{\text{models}} = 5, 7$ (Figure 5c). On SL Animals, while certain random combinations surpassed TreeSNNs at specific ensemble sizes, our approach generally ranked near the upper bound of random performance (Figure 5b). These findings demonstrate that TreeSNNs effectively selects $N_f$ values, yielding ensembles with overall stronger performance.

### 4.5 IMPACT OF NUMBER OF MODELS

We next examine the effect of varying the number of models ($N_{\text{models}}$), as this choice directly impacts the temporal variability that can be captured, and thus the final TreeSNN ensemble accuracy. Figure 6 shows the relationship between $N_{\text{models}}$ and ensemble accuracy, alongside individual model performance. On DVS-Gesture, accuracy improved consistently as $N_{\text{models}}$ increased, before saturating beyond a certain ensemble size, from 89.24% with a single model to 95.83% with nine-model ensembles. For SL-Animals DVS, the best performance was achieved at $N_{\text{models}} = 9$ with $N_f = 10, 15, 20, 30, 35, 40, 45, 95, 100$, while a slight dip occurred at $N_{\text{models}} = 7$, suggesting less consistent ensemble benefits for this dataset. On THU$^{\text{E-ACT}}$-50 CHL, accuracy increased from 67.28% to 74.08% with $N_f = 10, 15, 30, 40, 100$, but declines with more number of model ensembles. This is likely attributed to the intra-class variability and inter-class similarity of this dataset, and also highlights the importance of calibrating $N_{models}$ to dataset characteristics.

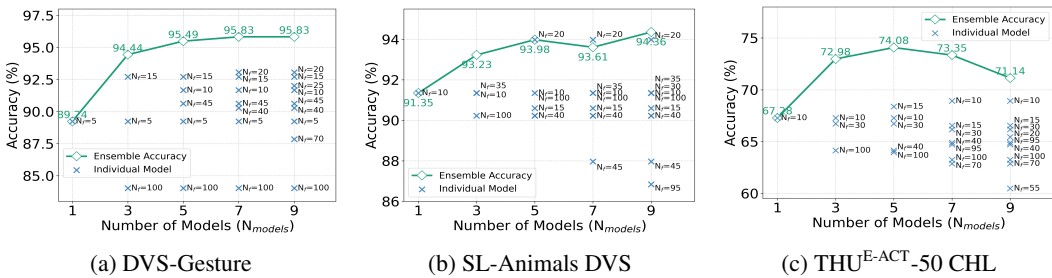

(a) DVS-Gesture      (b) SL-Animals DVS      (c) THU$^{\text{E-ACT}}$-50 CHL

Figure 6: Comparison of $N_{\text{models}}$ versus ensembling test accuracy for three datasets, including the performance of individual models.

Table 2: Test accuracy across $N_f$ for three neuromorphic action datasets.

| (a) DVS-Gesture | | (b) SL-Animals DVS | | (c) THU$^{\text{E-ACT}}$-50 CHL | |
|---|---|---|---|---|---|
| $N_f$ | Top-1 Acc. (%) | $N_f$ | Top-1 Acc. (%) | $N_f$ | Top 1 Acc. (%) |
| 5 | 89.24 (baseline) | 10 | 91.35 (baseline) | 10 | 67.28 (baseline) |
| 10 | 91.67 | 15 | 90.6 | 15 | 68.38 |
| 15 | 92.71 | 20 | 93.98 | 30 | 66.73 |
| 45 | 90.62 | 40 | 90.23 | 45 | 63.97 |
| 100 | 84.02 | 100 | 90.23 | 100 | 64.15 |
| **Ensemble** | **95.49 (+6.25%)** | **Ensemble** | **93.98 (+2.63%)** | **Ensemble** | **74.08 (+6.8%)** |

## 4.6 VISUALISATION OF ENSEMBLED MODEL PERFORMANCE

Table 2 summarizes the results of individual models and the ensemble across all datasets with $N_{\text{models}} = 5$, showing that ensembling consistently improves accuracy over individual models. To gain deeper insights into per-class behavior, we further analyzed predictions from selected classes of the challenging THU$^{\text{E-ACT}}$-50 CHL dataset, studying how models trained at different temporal resolutions complement one another. Figure 7 illustrates class-wise outcomes, where correct classifications fall inside the corresponding petal and misclassifications lie outside. It is evident that different $N_f$ values specialize in different classes. For instance, Class 1 achieves its best performance at $N_f = 30$, Class 40 benefits most from $N_f = 100$, while Class 25 is better captured at $N_f = 10$. These results demonstrate how ensembling allows TreeSNNs to leverage on the complementary class-specific accuracy of different models, resulting in improved overall performance.

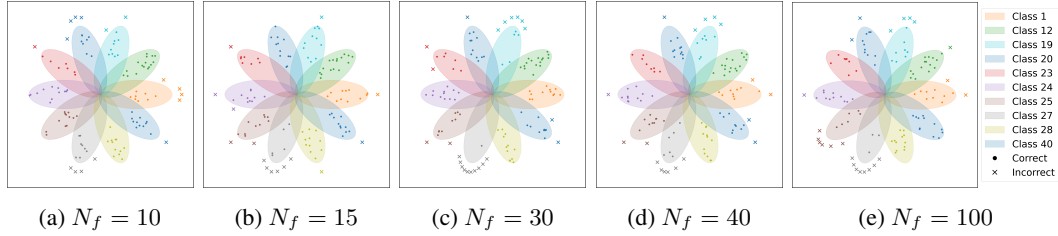

(a) $N_f = 10$     (b) $N_f = 15$     (c) $N_f = 30$     (d) $N_f = 40$     (e) $N_f = 100$

Figure 7: Class-wise prediction plots for different numbers of frames ($N_f$) on the THU$^{\text{E-ACT}}$-50 CHL dataset.

## 5 DISCUSSION

To better understand the trade-off between performance improvement and energy consumption, we computed the average synaptic operations (SOPs) per inference of SpikeResformer-Ti model across all datasets and compared the TreeSNNs ($N_{\text{models}} = 5$) with the corresponding single-model baselines. Synaptic operations provide a hardware-agnostic estimate of computational cost and are commonly used to approximate energy consumption in SNNs.

Table 3: Performance and energy comparison between single model (SpikeResformer-Ti) and TreeSNN ensembles.

| Dataset | Single Model | | TreeSNNs | |
| --- | --- | --- | --- | --- |
| | Acc. (%) | Energy (G) | Acc. (%) | Energy (G) |
| DVS-Gesture | 89.24 | 0.37524 | 95.49 (**+6.25**) | 1.93858 (**5.16**×) |
| SL-Animals | 91.35 | 0.90299 | 93.98 (**+2.63**) | 4.87474 (**5.39**×) |
| Thu50 | 67.28 | 0.94477 | 74.08 (**+6.80**) | 4.15103 (**4.39**×) |

Our results show that using an ensemble substantially increases computational demand, between $4\times$ and $6\times$ more SOPs, yet provides significant accuracy gains across datasets. On DVS-Gesture, the average SOP count increased by $5.16\times$ relative to the single baseline model, resulting in a $6.25\%$ improvement in accuracy. Similarly, for the other datasets, we observe SOP increases of $5.39\times$ and $4.39\times$, accompanied by accuracy gains of $2.63\%$ and $6.8\%$, respectively. These results highlight a consistent trade-off: energy consumption increases with ensemble size, but so does classification performance.

## 6 CONCLUSION AND FUTURE WORKS

We demonstrate that different temporal dynamics of event streams are best captured at distinct resolutions. To exploit this property, we propose TreeSNN, an SNN ensemble framework where multiple models operate on event frames at varying temporal resolutions. Central to our approach is the selection of these resolutions: we introduce the Fano factor to quantify such temporal dynamics and design an algorithm that identifies the set of bin choices that maximizes inter-class temporal differences. Experimental results confirm that incorporating such temporal diversity substantially boosts the performance of ensembled SNNs. We believe that this work will encourage further exploration of multi-temporal representation of events in SNN-based neural processing.

While our TreeSNN framework has demonstrated notable performance gains, several avenues remain open for further exploration. One promising direction is to reduce inference energy consumption. To tackle this, we can explore incorporating model selection based on the input sample, which will allow us to retain accuracy while reducing energy consumption. Another potential direction is to leverage the membrane potential dynamics of SNNs to estimate model confidence, a strategy particularly well suited to temporal ensembling where individual models contribute complementary strengths.

## USE OF LLMS

We have used LLMs for polishing and correcting grammar while writing the paper. We have also used LLM based coding assistants as support in tasks such as plotting results.

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
