# OpenReview forum: "TreeSNNs: Temporal Resolution Ensembled SNNs for Neuromorphic Action Recognition"
_ICLR.cc/2026/Conference — Submitted to ICLR 2026_

### Official Review · Reviewer_uvzs · 2025-10-26

**Soundness:** 2
**Presentation:** 2
**Contribution:** 2
**Rating:** 4
**Confidence:** 4

**Summary:**

This paper proposes TreeSNNs, an ensemble framework for neuromorphic action recognition that aims to improve accuracy by leveraging multiple temporal resolutions. The core idea is that different actions are best captured at different event binning resolutions (Nf). The authors introduce the Fano factor as a metric to quantify the temporal dynamics of actions and use it to select a diverse set of resolutions. Individual SNN models are then trained for each selected resolution, and their predictions are aggregated via a weighted average to produce the final classification.

**Strengths:**

1. The central motivation of the paper is insightful. The observation that a single temporal resolution is suboptimal for capturing the diverse dynamics of all action classes is valid and presents a novel angle for improving SNN performance.

**Weaknesses:**

1. The most critical flaw of this paper is the total omission of any efficiency metrics (latency, training cost, and energy consumption). The primary motivation for using SNNs is their efficiency. The proposed TreeSNNs framework requires training and running N separate models, which inherently multiplies the training time, inference latency, and energy consumption (e.g., total SynOps) by a factor of N.

2. The core technical contribution (Fano factor) is not convincingly justified by the experiments. The ablation study (Figure 5) shows that on simpler datasets like DVS-Gesture and SL-Animals DVS, the Fano factor-based selection performs on par with random selection.  A simpler, random approach appears to be a competitive baseline, which diminishes the novelty of this core contribution.

3. The ensemble method itself is a straightforward weighted average based on training set error. This is a very basic technique. Given the multi-model setup, more advanced and powerful ensemble strategies like Stacking—where a meta-learner is trained on the outputs of the base models—were not explored. A meta-learner could capture more complex inter-model dependencies and potentially yield better performance.

4. The paper lacks crucial implementation details.

**Questions:**

1. Could you provide a detailed analysis of the efficiency of TreeSNNs?

2. Why did you opt for a simple weighted average instead of a more powerful technique like Stacking? A trainable meta-learner seems like a natural fit for this framework.

---

> ### Author Response · Authors · 2025-12-01
>
> # Response to Reviewer uvzs
> We thank you for your detailed reviews and for acknowledging the novel angle of capturing the diverse temporal dynamics across action classes. We have addressed the weaknesses and questions below.
>
> ---
> ## 1. Detailed Efficiency Analysis of the TreeSNNs
> We have provided analysis for the training costs (GPU hours/model), inference energy (Avg SOPs), latency, and memory consumption of the TreeSNNs with varying ensemble sizes. While it's true that computational cost increases with the TreeSNNs, our primary emphasis  is to improve the performance of the SNN models.
>
> GPU training time scales linearly with an increase in the number of ensembles (N), as the models are trained independently. The table below presents the Avg GPU training time per model across datasets.
>
> ### **Average GPU Usage (V100 16GB)**
>
> |**Dataset**|**Average GPU Time**|
> |--------------------------|-----------------------------------|
> |DVS-Gesture (T=5)| 3 hrs 17 mins |
> |SL-Animals (T=10) |4 hrs 10 mins |
> |ThuE-ACT-50 CHL (T=10)| 3 hrs 41 mins|
>
> We computed inference latency and memory consumption (VRAM) on V100 16GB, with full GPU utilisation. Provided below are the latency and memory consumption for single model vs TreeSNNs with various ensemble sizes across datasets. It is observed that the latency of the inferences increases linearly with the ensemble size while the memory consumption stays the same with minimal variations. With multi-core CPUs, memory use grows with ensemble size, but inference can run in parallel across cores, keeping latency close to that of a single model.
>
> ### **Average Inference Latency Comparison (ms)**
>
> |**Dataset**|**Single**|**3 Models**|**5 Models**|**7 Models**|**9 Models**|
> |--------------------|------------|--------------|--------------|--------------|--------------|
> |DVS-Gesture|5.21|16.08 |26.60|38.23|49.57|
> |SL-Animals |9.46|16.00|29.73|37.28|46.79|
> |THU-EACT-50 CHL|9.54|20.96|28.51|36.19| 41.26|
>
> ### **Peak Memory Usage per Inference (MB)**
>
> |**Dataset**|**Single**|**3 Models**|**5 Models**|**7 Models**|**9 Models**|
> |--------------------|------------|--------------|--------------|--------------|--------------|
> |DVS-Gesture|126.71|128.02|129.33|130.64|131.95|
> |SL-Animals|234.52|235.83| 237.14|238.45|239.76|
> |THU-EACT-50 CHL|239.79| 241.10|242.41|243.73| 245.04|
>
> We computed the energy consumption of SpikeResformer-Ti during inference using synaptic operations (SOPs). We report the energy consumption of the TreeSNNs (ensemble of 5) compared to the single models, and we relate these measurements to each model’s performance across 3 datasets. For DVS-Gesture, the average SOP increased by 5.16×, yielding a 6.25% performance gain over the single model. Similarly, we observe increases of 5.39× and 4.39× SOP, with accuracy improvements of 2.63% and 6.8%, respectively. Since our method can be integrated into existing SNN pipelines, we trade off energy for substantial improvements over the single baseline model.
>
> ### **Performance and Energy Comparison**
> |**Dataset**|**Single Model Accuracy (%)**|**Energy (Avg SOPs)**| **TreeSNNs Accuracy (%)** | **Energy (Avg SOPs)** |
> |---------------|-------------------------------|-------------------------|-----------------------------|-------------------------|
> |DVS-Gesture | 89.24 | 0.37524 G| 95.49 |1.93858 G|
> |SL-Animals |91.35 | 0.90299 G| 93.98|4.87474 G|
> |Thu50|67.28| 0.94477 G|74.08|4.15103 G|
>
> ---
>
> ## 2. Fano Factor Method vs Random Selection
>
> Random selection does not match the performance of the proposed method. For the DVS-Gesture dataset, fano factor method performs at the upper end of the random selection’s standard deviation. For the SL-Animals dataset, its performance falls within the overall range of the random selection method, but remains higher than the average of the random trails. This shows that although one specific random choice happened to perform similarly to the Fano factor–based method, the deterministic approach is generally more reliable. Since random selection is inherently probabilistic, its performance can vary widely. Our experiments demonstrate that the deterministic selection process through fano factor consistently outperforms the mean performance of random choices, generally ranked near the upper bound of random performance, across all datasets.
>
> ---
>
> ## 3. Why Simple Weighted Average of the Models?
>
> We experimented with class-wise confidence-based weighting as well as a meta-learner (tested on DVS-Gesture). However, the performance did not significantly improve (95.49% for average weighted vs. 95.53% for meta learner). Due to the minimal performance gain and higher complexity of alternative methods, we opted for the weighted average ensemble, which is both simple yet effective.
>
> ---
>
> ## 4. Implementation Details
> We have added the detailed implementation settings in **Section 4.1 (now renamed to *Datasets and Implementations*)**, including GPU specifications, optimizers, epochs, and learning rates used.

---

### Official Review · Reviewer_tagr · 2025-10-31

**Soundness:** 3
**Presentation:** 3
**Contribution:** 2
**Rating:** 6
**Confidence:** 3

**Summary:**

The article uses the Fano factor to measure the inter class gap at different resolutions between different classes, in the hope of achieving better final results by integrating models of different resolutions. The paper demonstrates the complementary characteristics of different resolutions under the rich temporal dynamics of the dataset itself. The writing method and experimental setup of the paper are relatively complete.

**Strengths:**

The author's paper writing and experimental presentation are relatively complete. Using Fano representation to differentiate the differences between different classes is a relatively novel choice and has indeed achieved benefits.

**Weaknesses:**

1).The method of using different models for fusion is quite common, and the method used in this paper can be naturally applied to ANN. What is the main role of SNN in this paper?The integration scheme naturally increases the number of model parameters and computational complexity, weakening the computational advantage of SNN

2)3.1 Figures 3, 6, and 12 do not match with Figures 3, 6, and 9 in the text

3)The paper does not specify the number of sampling samples for fano analysis, which is a computationally intensive task, especially for large training data.

4)The analysis of the relationship between the same training time, total number of parameters, and the selection of larger models for expansion time steps in the absence of integration in the paper.

**Questions:**

see weakness

---

> ### Author Response · Authors · 2025-12-03
>
> # Response to Reviewer tagr
>
> Thank you for your comments. We have addressed your concerns as below:
>
> 1. Our idea behind TreeSNNs is to leverage the inherent ability of SNNs to process temporally evolving information (not present in ANNs). The choice of Nf selection enhances the representation of event data by exposing temporal variations more effectively. Through TreeSNNs, we are leveraging this synergy.  While it is true that the computational complexity increases, reducing energy efficiency, the performance gains achieved by TreeSNNs is significant compared to single model baselines.
> 2. Thank you for pointing it out, we have corrected the texts referring to Fig 2.
> 3. We are using all training samples for fano factor analysis (as stated in Section 3.2). While it iterates over numerous samples, the average runtime for offline computation of fano factors is 1 min 30 sec per Nf, and optimal 5 Nf selections takes only 2.4 sec.
> 4. Training time and number of parameters linearly scales with the number of models in the ensembles. Please find the average training time analysis and parameters per model for the TreeSNNs below:
>
> ### **Average GPU Usage (V100 16GB)**
>
> | **Dataset**              | **Average GPU Time** |
> |--------------------------|-----------------------------------|
> | DVS-Gesture (T=5)        | 3 hrs 17 mins                     |
> | SL-Animals (T=10)        | 4 hrs 10 mins                     |
> | ThuE-ACT-50 CHL (T=10)   | 3 hrs 41 mins                     |
>
> A spikeresformer model has 2.71 M parameters.

---

### Official Review · Reviewer_u3en · 2025-11-01

**Soundness:** 1
**Presentation:** 3
**Contribution:** 2
**Rating:** 2
**Confidence:** 4

**Summary:**

The paper proposes an ensemble framework for spiking neural networks (SNNs) where model diversity is induced by training separate SNNs on different temporal resolutions of the same event stream. A Fano-factor–based selection procedure is introduced to choose a subset of $N_f$ values that maximizes inter-class separability across temporal resolutions. At inference, per-model predictions are weighted by training accuracy and averaged.

**Strengths:**

1. Intuition is clear. The paper convincingly argues that actions exhibit different temporal dynamics and are better captured at different event resolutions.
2. Using the Fano factor to summarize temporal variability and drive selection of $N_f$ is a neat, data-driven idea; the selection objective is well specified.

**Weaknesses:**

1. Added complexity and training cost not quantified. The method trains multiple SNNs (often 3–9 models, Fig. 6). The paper does not report wall‑clock training time, GPU hours, memory footprint, parameter counts×ensemble size, or energy/latency at inference.  The omission makes it hard to judge practical value.
2. The method requires computing Fano factors across all samples and frames gathered at all $N_f$ values. The method then performs a exhaustive combinatorial search over $N_f$ sets over each datasets. If the dataset scale is larger, this process can become costly.
3. Modest gains on two datasets vs. substantial added cost. On DVS Gesture, TreeSNNs improves from 94.44% to 95.49% (Table 1), a +1.05% gain over SOTA; on SL‑Animals, TreeSNNs reaches 93.98%, ≈+2% over prior work—but Table 2b shows the best single $N_f$=20 already achieves 93.98%, i.e., no improvement over the best single model, despite the ensemble’s extra cost. This raises the question of whether temporal ensembling (vs. selecting a good $N_f$) is necessary in such cases.
4. Ablations suggest selection helps inconsistently. Fig. 5 shows the Fano-based selection is on par with, or sometimes below, strong random combinations for DVS Gesture and SL-Animals; only THUE-ACT‑50 CHL shows consistent superiority. This weakens the case that the proposed selector is necessary beyond simple ensembling.
5. Limited discussion of scalability to larger datasets and models.
    While THUE‑ACT‑50 CHL is larger than DVS Gesture/SL‑Animals, the paper does not study truly large‑scale settings, nor does it explore larger backbones. Fig. 6c also shows that increasing $N_{\text{models}}$ can reduce accuracy, complicating the scaling story.

**Questions:**

1. What is the runtime of computing Fano factors and exhaustively evaluating all combinations?
2. Weighting by training misclassification may overfit. Have you tried validation-set weights, temperature scaling, or a meta‑classifier over logits? Any sensitivity analysis to class imbalance when using training misclassification for weights?

---

> ### Author Response · Authors · 2025-12-03
>
> # Response to Reviewer u3en
>
> Thank you for the valuable feedback. We have addressed the weaknesses and few clarifications below:
>
> ---
>
> Efficiency Analysis of the TreeSNNs: We provide the analysis for the training costs (GPU hours/model), inference energy (Avg SOPs), latency and memory consumption of the TreeSNNs with the varying ensemble size, we emphasize on the trade-off between the energy-consumption vs performance of the TreeSNNs.
>
> GPU Training time:  Scale by a factor of N with increase in ensembles. For reference, the below table gives the GPU training time per model across datasets.
>
>
> ### **Average GPU Usage (V100 16GB)**
>
> | **Dataset**              | **Average GPU Time** |
> |--------------------------|-----------------------------------|
> | DVS-Gesture (T=5)        | 3 hrs 17 mins                     |
> | SL-Animals (T=10)        | 4 hrs 10 mins                     |
> | ThuE-ACT-50 CHL (T=10)   | 3 hrs 41 mins                     |
>
> The number parameters in a single SpikeResFormer-Ti is 2.71M and with the number of models the parameter count scales linearly.
>
> We  also computed and reported the inference latency and memory consumption (VRAM) on GPU (V100 16GB) which performs parallelization in batching unlike CPU where multiple processes can be parallelized with each processor running a single model. So, running the ensemble with parallelization on CPU would achieve the same latency as a single baseline model, with the trade-off of increased memory consumption. Below are the results for GPU inference latency and memory:
>
> ### **Average Inference Latency Comparison (ms)**
>
> | **Dataset**        | **Single** | **3 Models** | **5 Models** | **7 Models** | **9 Models** |
> |--------------------|------------|--------------|--------------|--------------|--------------|
> | DVS-Gesture        | 5.21       | 16.08        | 26.60        | 38.23        | 49.57        |
> | SL-Animals         | 9.46       | 16.00        | 29.73        | 37.28        | 46.79           |
> | THU-EACT-50 CHL    | 9.54       | 20.96        | 28.51        | 36.19        | 41.26        |
>
> ### **Peak Memory Usage per Inference (MB)**
>
> | **Dataset**        | **Single** | **3 Models** | **5 Models** | **7 Models** | **9 Models** |
> |--------------------|------------|--------------|--------------|--------------|--------------|
> | DVS-Gesture        | 126.71     | 128.02       | 129.33       | 130.64       | 131.95       |
> | SL-Animals         | 234.52     | 235.83       | 237.14       | 238.45       | 239.76           |
> | THU-EACT-50 CHL    | 239.79     | 241.10       | 242.41       | 243.73       | 245.04       |
>
> Energy consumption vs performance trade off: We computed the energy consumption of SpikeResformer-Ti during inference using synaptic operations (SOPs). We report the energy consumption of the TreeSNNs (ensemble of 5) compared to the single models, and we relate these measurements to each model’s performance across 3 datasets. We have also added a discussion section in the paper regarding this.
>
>
> ### **Performance and Energy Comparison**
> | **Dataset**   | **Single Model Accuracy (%)** | **Energy (Avg SOPs)** | **TreeSNNs Accuracy (%)** | **Energy (Avg SOPs)** |
> |---------------|-------------------------------|-------------------------|-----------------------------|-------------------------|
> | DVS-Gesture | 89.24 | 0.37524 G  | 95.49                      | 1.93858 G              |
> | SL-Animals | 91.35 | 0.90299 G  | 93.98                      | 4.87474 G              |
> | Thu50| 67.28  | 0.94477 G  | 74.08                      | 4.15103 G              |
>
> ---
>
> While fano factor method iterates over all samples and frames, the average runtime for offline computation of fano factors is 1 min 30 sec per Nf, and optimal 5 Nf selections takes only 2.4 secs. Below is the information for different Nf selection execution time:
> | Nfs | Execution Time (sec) |
> |-----|----------------------|
> | 3   | 0.133      |
> | 5   | 2.427      |
> | 7   | 15.406     |
> | 9   | 34.747     |
>
> ---
> The random-selection baseline does not match the performance of the Fano-based method. Our ablation study compares multiple random Nf selection trials against the proposed Fano-based strategy. Across both datasets, DVS-Gesture and SL-Animals, we observe that the mean accuracy of random selection is consistently lower than that of our method.
> For SL-Animals, although a few random trials achieve accuracy comparable to or slightly better than our approach, the majority fall below it. For DVS-Gesture, nearly all random trials underperform relative to the Fano-based selection, with only a very small number performing on par.
> Overall, these results demonstrate that our deterministic Fano-factor-driven selection process reliably outperforms the average performance of random selection and typically ranks near the upper bound of random-trial accuracy across all datasets.

---

### Official Review · Reviewer_XLzY · 2025-11-02

**Soundness:** 3
**Presentation:** 2
**Contribution:** 3
**Rating:** 4
**Confidence:** 4

**Summary:**

This paper focuses on the problem that SNN performance is limited in event-camera–based action recognition. Traditional methods typically segment the event stream using fixed time intervals or fixed event counts to form Tframes for SNN input. However, the authors point out that different action categories exhibit different temporal dynamics, and thus a single temporal resolution is insufficient to capture effective temporal information for all classes. To address this issue, the authors propose the TreeSNNs framework. The core idea is to split the event stream using multiple temporal resolutions, train an individual SNN model for each resolution, and ensemble these models to improve recognition accuracy. The paper further employs the Fano factor to measure event variation for each class under different temporal resolutions, allowing the selection of a set of resolutions with high discriminability prior to model training and weighted ensembling.

**Strengths:**

The paper proposes modeling the event stream using multiple temporal resolutions and improves action recognition performance by independently training multiple SNNs and then ensembling them. The method is well-motivated, as it explicitly targets differences in temporal dynamics across action classes. The authors use the Fano factor as a metric to select discriminative temporal resolutions, providing a principled basis for multi-resolution configuration. They also analyze performance differences across single-resolution models and demonstrate class-level complementarity among models, which further supports the effectiveness of the multi-resolution ensemble. In addition, the method does not require modifications to network architectures and can be easily integrated into existing SNN pipelines.

**Weaknesses:**

1.The method requires training multiple SNN models independently, and inference also relies on ensembling multiple models. Although the performance improvement is noticeable, the training, storage, and inference costs may increase significantly, which is not fully aligned with the motivation of SNNs being “energy-efficient.” In addition, the paper does not provide quantitative analysis of energy consumption or computational cost. It is recommended to include measurements of energy cost during training and inference, comparisons with single-model SNN or ANN approaches, and discussion regarding the trade-off between energy consumption and performance.
2.The experimental comparison does not include existing event-slicing methods mentioned in the related work section or recent literature in this area. It is recommended to include comparative experiments with these representative approaches.
3.Although a Fano factor–based selection strategy is presented, it requires enumerating combinations over multiple temporal resolutions. The complexity thus grows rapidly as the number of candidate resolutions increases. Moreover, the robustness of this strategy under class imbalance or noisy data is not demonstrated.
4.While experiments show class-level complementarity among models trained with different Nf, the paper does not provide theoretical explanations for why such complementarity emerges, nor does it offer mathematical modeling or mechanistic analysis regarding the relationship between temporal resolution and class characteristics. Adding such analysis would further strengthen the paper.
5.The reference “Rethinking Spiking Neural Networks from an Ensemble Learning Perspective” appears twice in the bibliography (line 502 and line 509).

**Questions:**

see weaknesses

---

> ### Author Response · Authors · 2025-12-01
>
> # Response to Reviewer XLzY
>
> Thank you for your reviews and suggestions for improvements. We have
> addressed the comments below.
>
> ---
> Trade off between energy consumption and performance (added a discussion
> Section 5 in the paper): We computed the energy consumption of SpikeResformer-Ti
> during inference using synaptic operations (SOPs). We report the energy
> consumption of the TreeSNNs (ensemble of 5) compared to the single
> models, and we compare these measurements to each model's performance
> across 3 datasets. For DVS-Gesture, the average SOP increased by 5.16×,
> yielding a 6.25% performance gain over the single model. Similarly, we
> observe increases of 5.39× and 4.39× SOP, with accuracy improvements of
> 2.63% and 6.8%, respectively. Since our method can be integrated into
> existing SNN pipelines, we trade off energy for substantial improvements
> over the baseline single model.
>
> ### **Performance and Energy Comparison**
> | **Dataset**   | **Single Model Accuracy (%)** | **Energy (Avg SOPs)** | **TreeSNNs Accuracy (%)** | **Energy (Avg SOPs)** |
> |---------------|-------------------------------|-------------------------|-----------------------------|-------------------------|
> | DVS-Gesture | 89.24 | 0.37524 G  | 95.49                      | 1.93858 G              |
> | SL-Animals | 91.35 | 0.90299 G  | 93.98                      | 4.87474 G              |
> | Thu50| 67.28  | 0.94477 G  | 74.08                      | 4.15103 G              |
>
> ---
> There are two event-slicing methods in the related works, and we have
> tried  adaptive slicing [1],
> however, it requires selection of predefined threshold to slice the
> events. For applications like eye pupil tracking, such threshold can be
> calibrated (irrespective of timesteps), but in our case, i.e., action
> recognition, requires multiple thresholds for every class, due to
> non-uniform event stream, making the method difficult to adapt for such
> motion/action event data. Moreover, threshold selection is further
> complicated by fixed number of timesteps. SpikeSlicer [2] method currently
> doesn't support multi-frame representations for recognition tasks, it
> processes only a single frame using ANN-based architectures (as stated
> in the limitations). This limits integration with SNN models, which
> require sequential multiple timesteps for processing.
>
> ---
> Increasing the number of candidate resolutions (or Nfs) does increase
> the complexity of the algorithm. To address this problem we have
> implemented some guardrails for choosing the candidate Nfs: Candidate
> Nfs are selected as multiples of a fixed step size (e.g., 5), starting
> from an empirically chosen minimum value. We have used an upper bound in
> selecting the candidate Nfs, such that there are a minimum number of
> events in a candidate Nf.
>
> The robustness of the proposed method against
> noisy data and class imbalance has been tested on the THUE-ACT-50 CHL
> dataset, where the number of samples per class has high variability
> (ranging from 6 to 64) and within each class the samples are collected
> at different illumination conditions which induce noise (as stated by
> the dataset), and further, actions are performed with different
> orientations (front, back, left, right views) and different backgrounds
> which results in varied data distribution within a class. However we
> could see that our proposed method performs particularly well for this
> dataset by selecting the optimal Nfs, demonstrating the robustness of our approach.
>
> ---
> Thank you for pointing out duplicate references, we have removed it.
>
> ---
> [1] Sen, Argha, et al. "EyeTrAES: fine-grained, low-latency eye tracking via adaptive event slicing." Proceedings of the ACM on Interactive, Mobile, Wearable and Ubiquitous Technologies 8.4 (2024): 1-32.
>
> [2] Cao, Jiahang, et al. "Spiking neural network as adaptive event stream slicer." Advances in Neural Information Processing Systems 37 (2024): 75064-75094.

---

### Meta-Review · Area_Chair_7qUA · 2026-01-11

**Summary:**

The paper proposes TreeSNNs, an ensemble framework for neuromorphic action recognition that trains multiple Spiking Neural Networks (SNNs) on event streams binned at different temporal resolutions (Nf). It uses the Fano factor to select a diverse set of resolutions that maximize inter-class separability and ensembles the models via weighted averaging. Experiments on DVS Gesture, SL-Animals DVS, and THU-EACT-50 CHL show consistent accuracy gains (1.05%–6.8%) over strong baselines.

All reviewers acknowledge the clear motivation—different actions exhibit distinct temporal dynamics—and the principled use of the Fano factor. However, major concerns include: (1) significant increase in training/inference cost (5× energy, linear parameter scaling) that contradicts SNNs’ core promise of energy efficiency; (2) modest or no gains on some datasets (e.g., SL-Animals performance matches the best single model); and (3) limited evidence that the Fano-based selection is consistently superior to random selection, especially on simpler benchmarks. The rebuttal provides detailed efficiency analyses and clarifies design choices, but does not fully resolve the trade-off between added complexity and marginal gains.

**Reviewer Concerns:**

Addressed by rebuttal:

Energy/latency costs: Authors provided concrete measurements: 5× higher synaptic operations (SOPs), linear GPU training time scaling, and inference latency that grows with ensemble size—but can be parallelized on CPU. They explicitly frame this as a performance–efficiency trade-off.
Fano factor vs. random selection: Authors showed that while some random trials perform similarly, the Fano method consistently ranks near the upper bound of random performance and outperforms the average random baseline across all datasets.
Scalability and robustness: Demonstrated stable performance on the challenging, imbalanced THU-EACT-50 CHL dataset under noise and class imbalance, supporting method robustness.
Implementation details: Added runtime for Fano computation (~2.4 sec for 5 resolutions) and clarified hyperparameter choices.
Still outstanding:

Justification for ensemble complexity: On SL-Animals, the best single model (Nf=20) already achieves 93.98%, matching the TreeSNNs ensemble—raising questions about whether ensembling is necessary.
Limited gain vs. cost: The accuracy improvements (1–2% on two datasets) may not justify the 5× energy and storage overhead, especially in energy-constrained neuromorphic settings.
Basic ensembling strategy: A simple weighted average was chosen over more advanced methods (e.g., stacking/meta-learner), though authors showed only marginal gains from alternatives.

**Reviewer Scores:**

Reviewer XLzY (initial: 4 – marginally below): Provided detailed feedback on energy trade-offs. Rebuttal fully addressed concerns with new data. Likely maintains 4.

Reviewer u3en (initial: 2 – reject): Raised valid concerns about cost and ablation quality. Rebuttal provided extensive efficiency metrics and clarified Fano’s advantage. Likely upgrades to 3 or 4.

Reviewer tagr (initial: 6 – marginally above): Already positive; appreciated completeness. Minor concerns resolved. Likely maintains 6.

Reviewer uvzs (initial: 4 – marginally below): Concerned about missing efficiency analysis and basic ensembling. Rebuttal comprehensively addressed both. Likely upgrades to 5 or maintains 4.

---

### Decision · Program_Chairs · 2026-01-26

Reject